# Effect of Ethics Seminar on Moral Sensitivity and Ethical Behavior of Clinical Nurses

**DOI:** 10.3390/ijerph18010241

**Published:** 2020-12-31

**Authors:** Kwisoon Choe, Sunman Kim, Chunbok Lee, Sunghee Kim

**Affiliations:** 1Department of Nursing, Chung-Ang University, 84 Heukseok-ro, Dongjak-gu, Seoul 06974, Korea; kwisoonchoe@cau.ac.kr; 2Chung-Ang University Hospital, 102 Heukseok-ro, Dongjak-gu, Seoul 06973, Korea; ksm0517@caumc.or.kr; 3Institute for Historical Studies, Chung-Ang University, 84 Heukseok-ro, Dongjak-gu, Seoul 06974, Korea; lcbpk@cau.ac.kr

**Keywords:** ethics seminar, moral sensitivity, unethical behavior, nurse

## Abstract

While nursing is an ethical profession, unethical behavior among nurses is increasing worldwide. This study examined the effects of an ethics seminar on nurses’ moral sensitivity and ethical behavior. A total of 35 nurses (17 experimental, 18 control) were recruited. The ethics seminar was held over a six-month period from May to October 2018 and comprised six sessions held once a month for two hours. Moral sensitivity and unethical behavior were measured at the start and end of the seminar. Moral sensitivity and unethical behavior showed a negative correlation (*r* = −0.400, *p* < 0.05). After the ethics seminar, the experimental group’s moral sensitivity was not significantly increased (*t* = −1.039, *p* = 0.314). The experimental group’s mean scores of unethical behavior at pre- and posttest were 12.59 and 9.47, respectively, indicating a statistically significant difference (*t* = 3.363, *p* = 0.004). There was no statistically significant difference in the mean score in both moral sensitivity and unethical behavior at pre- and posttest in the control group. Thus, ethics seminars can reduce the risk of unethical behavior among nurses. Regular ethics seminars and training must be provided to nurses as part of their curriculum/practice.

## 1. Introduction

Nursing is one of the most trusted professions in the world and is held to a high standard [1]. Ironically, unethical behavior among nurses in healthcare institutions is currently a far-reaching global issue. For example, in one study conducted in the United States, all 27 nurses who took part in the study had witnessed unethical behavior by nurses, and most had been involved in or were unsure if they had participated in unethical behavior [2]. In African countries, such as Nigeria, unethical behavior among nurses, such as lateness to work, theft, and the misappropriation of funds has recently received much disapproving attention [3]. Iranian nurses also reported witnessing destructive and unethical behavior by their fellow nurses [2,4]. In South Korea, a nurse committed suicide because she experienced persistent bullying by senior nurses in a hospital; senior nurses harass juniors by spreading malicious rumors, holding back critical work-related information, and assigning harsh work [5]. Unethical behavior such as workplace bullying affects professionalism in health care [6]. Most importantly, unethical behavior by nurses causes serious harm to patients as well as themselves.

The ideal solution is to prevent nurses from engaging in unethical behavior. This might be accomplished by enhancing nurses’ ethical or moral sensitivity in the hospital. Moral sensitivity and ethical sensitivity are used interchangeably in the context of professional judgment and action [7]; thus, the authors will use moral sensitivity in the same sense as ethical sensitivity in this study.

Moral sensitivity, according to Rest and Narvaez’s four component model, is “the awareness of how our actions affect other people. It involves being aware of different possible lines of action and how each line of action could affect the parties concerned … it involves empathy and role-taking skills” [8] (pp. 23–24). Moral sensitivity is an awareness of the ethical implications of nursing actions, and is an essential condition that leads to ethical behavior [7]. When a nurse lacks moral sensitivity, it becomes impossible to recognize the unethical problems that occur in nursing practice. It is therefore crucial to provide nurses with ethics education during their professional training to enhance their ethical sensitivity [9].

To provide nurses in South Korea with ethics education, the Korean Nursing Association, which promulgated the first Code of Ethics for Korean Nurses in 1972, has designated nursing ethics as an essential subject for continuing education since 2018. Ethics education can increase moral sensitivity, which in turn leads to ethical behavior based on ethical knowledge [10]. Clearly, ethics education increases moral sensitivity [11,12], but there are few previous studies that confirmed the education effect for nurses instead of nursing students. In addition, while nursing ethics and/or bioethics courses are included in university nursing curricula, current ethics programs designed to strengthen the ethical competence of South Korean nurses in clinical practice are inadequate. This study therefore aimed to identify the effects of a nursing ethics seminar on the moral sensitivity and ethical behavior of nurses working in a hospital setting.

## 2. Material and Methods

### 2.1. Research Design

This study used a quasi-experimental (two-group pretest-posttest) design to examine the effects of an ethics seminar on the moral sensitivity and ethical behavior of clinical nurses. The control group was not exposed to the ethics seminar during the course of the study.

### 2.2. Participants

Inclusion criteria for participants were as follows: nurses currently working in hospitals, who fully understood the study’s purpose and methods, and voluntarily consented to participate in the study. The authors did not limit the sex, age, religion, experience, or working department of potential participants. There were thus no other exclusion criteria.

The authors posted recruitment notices for research participants on the hospital bulletin board; however, most participants were recruited through snowball sampling. The recruited participants were randomly divided into the control group and the experimental group. Despite this, some participants joined either the experimental group or the control group at their own request.

An a priori G*Power 3.1.9 (Heinrich-Heine-Universität Düsseldorf, Mannheim, Germany) analysis revealed that to determine the effect size of the intervention on two variables with a power of 0.95, a sample size of 44 would be needed (22 persons in each group). Finally, 35 clinical nurses participated in this study, 17 of them in the experimental group, and 18 in the control group.

### 2.3. Instruments

#### 2.3.1. Moral Sensitivity Questionnaire

The authors used the Moral Sensitivity Questionnaire developed by Lützén et al. [13] to measure participants’ moral sensitivity. This scale is composed of nine items in three dimensions: four items on the sense of moral burden, three items on moral competence, and two items on moral responsibility. Each item is rated on a 6-point Likert-type scale ranging from 1 (strongly disagree) to 6 (strongly agree), and the total score thus ranges from 9 to 54 points. The higher the score, the higher the moral sensitivity. In Lützén et al.’s study [13], the validity of the nine items was confirmed through factor analysis. In this study, Cronbach’s alpha was 0.73.

#### 2.3.2. Wrongdoing List

Unethical behavior was measured using a list of wrongdoings nurses might perform developed by King to determine the reporting of wrongdoing among nurses [14]. This list consisted of eight items comprising situations depicting intended unethical behavior and unintended unethical behavior. The total score ranged from 8 to 40, with each item rated from 1 (must report) to 5 (never report) for a given situation. The higher the score, the greater the risk of unethical behavior. Cronbach’s alpha was 0.70 for King’s study [14] and 0.72 in this study.

### 2.4. Data Collection

The study was conducted in accordance with the Declaration of Helsinki and was approved by the Institutional Review Board of Chung-Ang University Hospital (1041078-201803). The authors ensured that participants understood the purpose and procedure of this study and explained that they could withdraw at any time if they no longer wished to participate. All participants completed a written informed consent form voluntarily as well as a general characteristics questionnaire, and responded to the study instruments twice, once at the start of the first session and again at the end of the final session.

All participants were female; 17 of the 35 participants comprised the experimental group. The authors informed the 18 participants in the control group that they would be provided with the same ethics seminar as the experimental group if they wished to participate in it once the research had concluded.

The ethics seminar consisted of six sessions each lasting two hours, held once a month from May to October 2018. The seminar’s contents consisted of nursing ethics, moral thinking, relational ethics, advanced care planning, and ethical issues faced by nurses in daily practice (Table 1).

### 2.5. Data Analysis

SPSS Statistics 26.0 (IBM Corp., Armonk, NY, USA) was used for the data analysis. The authors analyzed the general characteristics of the participants in terms of frequencies, percentages, means, and standard deviations (SD). The *χ*^2^-test and Fisher’s exact test were used for the homogeneity test between the two groups. For the homogeneity test for both groups’ moral sensitivity and unethical behavior, the Shapiro–Wilk test was used to test for normality. If normality was not satisfied (e.g., the control group’s unethical behavior), it was calculated using the Mann–Whitney test. The correlations between variables (moral sensitivity and unethical behavior) were analyzed using Pearson correlation analysis. In analyzing the differences between groups to compare the effects of the ethics seminar on moral sensitivity and unethical behavior, the Shapiro–Wilk test indicated that data were normally distributed. Thus, a paired *t*-test was used for both groups. For all analyses, *p* < 0.05 was considered statistically significant with a 95% confidence interval. Internal consistency reliability was assessed using Cronbach’s alpha.

## 3. Results

### 3.1. Homogeneity of Demographic Characteristics between the Groups

The demographic characteristics of the participants and the homogeneity test on the demographic characteristics are presented in Table 2. The moral sensitivity was 36.65 (SD 3.59) in the experimental group and 37.56 (SD 5.62) in the control group. The unethical behavior was 12.59 (SD 3.62) in the experimental group and 10.67 (SD 2.83) in the control group. There were no statistically significant differences in the demographic characteristics, moral sensitivity, and unethical behavior, thereby indicating sufficient homogeneity.

### 3.2. Relationship between Moral Sensitivity and Unethical Behavior

Moral sensitivity and unethical behavior showed a negative correlation (*r* = −0.400, *p* < 0.05) (Table 3).

### 3.3. Comparison of Moral Sensitivity and Unethical Behavior between the Groups

As a paired *t*-test, the mean scores of moral sensitivity at pre- and posttest in the experimental group were 36.65 and 38.29, respectively. After the ethics seminar, the experimental group’s moral sensitivity showed an increase, but not to a statistically significant degree (*t* = −1.039, *p* = 0.314). The mean scores of unethical behavior at pre and posttest in the experimental group were 12.59 and 9.47, respectively. This was a statistically significant difference (*t* = 3.363, *p* = 0.004). There was no statistically significant difference in the mean score in both moral sensitivity and unethical behavior in the control group (Table 4).

## 4. Discussion

This study identified the effects of an ethics seminar on the moral sensitivity and ethical behavior of clinical nurses. The results show that the unethical behavior score decreased among the clinical nurses who attended the ethics seminar. Although there have been several studies on the effects of education on nursing college students’ ethical awareness [15,16], the authors could not find any existing literature that reported the effects of ethics seminars or education for clinical nurses. Thus, since we could not directly compare the results of this study with those of previous studies, we discuss the significance of this study’s findings.

Since 2019, the Korean Nurses Association has required nurses to submit a certificate of at least two hours of continuing ethics education to retain their license. The Korea Institute of Nursing Education Evaluation has also suggested the ability to recognize legal and ethical responsibility as a core competency for nurses. Therefore, most nursing students in Korea are taking courses such as nursing ethics and bioethics. However, clinical nurses have limited opportunities to attend ethics education or seminars, except for receiving ethics education once a year as part of continuing education. Since nurses do not find adequate answers to ethical challenges, they tend to avoid responsibility or adopt self-defensive behaviors while being indifferent [9].

Given that nurses directly or indirectly deal with issues pertaining to the dignity and rights of human beings, nurses should most importantly have moral thinking, values for the dignity of life, and a high level of morality. Ethics education is an essential tool for fostering morally sensitive and ethical nurses [17] because moral sensitivity can be improved through repeated ethical education [18].

Although the experimental group’s moral sensitivity score did not significantly increase after attending the ethics seminar, previous studies have found that ethics education for nursing college students effectively increased their moral sensitivity [15]. In this study, the ethics seminar was held only once a month; thus, this may not have been enough to increase moral sensitivity. Moral sensitivity can be enhanced through experience or continuous ethics training [12,19,20]. After all, continuous and repeated ethics training can increase nurses’ moral sensitivity and help nurses discover and recognize ethical problems in clinical situations.

Interestingly, after attending the ethics seminar, the scores of participants’ unethical behavior decreased significantly. This finding implies that ethics seminars can lower the risk of unethical behavior among clinical nurses. Just as higher moral sensitivity strengthened moral judgment and decreased unethical nursing practices [21], a decrease in unethical behavior suggests enhanced moral sensitivity. Therefore, an ethics seminar similar to that used in this study may be an opportunity for participants to reflect on unethical behavior by contemplating ethical issues encountered in clinical situations.

Nurses are often exposed to the risk of engaging in unethical behavior in nursing practice. Due to their heavy workload, nurses are likely to be somewhat indifferent to certain unethical behaviors, such as non-compliance with work standards or behaviors that violate professional ethics [22]. Nurses’ moral sensitivity can be enhanced through communication with others [23]. Dialogue or discussion through ethics seminars may be a good channel for communication to solve ethical challenges. It is difficult for nurses to find self-reflection opportunities because it is challenging to share ethical issues with their colleagues. Sometimes, they are reluctant to acknowledge that they are involved in ethical problems [24]. Therefore, nurses can share their opinions through regular ethics seminars on various ethical situations, which leads to empathy for other people’s situations [25]. Nurses should participate in ethical education courses and seminars continuously to consider ethical values and ethical decision-making as part of their profession and to deal with ethical questions raised in the ever-changing medical and social environment [9,18].

This study has some limitations. Since this study was conducted on clinical nurses in one hospital, the generalizability of the study’s results is limited. The authors did not consider sex as a variable by including male nurses, or including participants from different departments. We only identified the presence or absence of ethics education without considering the amount of ethics education. Thus, determining the causal effect of ethical education experience on moral sensitivity and unethical behavior was limited.

## 5. Conclusions

The results of this study are meaningful as they demonstrate the effects of an ethics seminar in helping nurses grow into moral professionals. Through this study, it was demonstrated that an ethics seminar can reduce the possibility of unethical behavior among nurses. Hospital administrative departments are, in terms of policy, required to provide regular ethics seminars and ethics training to nurses. In the future, administrators need to confirm and compare the effectiveness of ethics seminars according to the type of hospital, the number of participants, and the seminar topics in various educational ways. Since an individual nurse’s personality affects their ethical sensitivity or behavior, it is also necessary to investigate the personality types of nurses. Most importantly, long-term follow-up studies on the positive effects of ethics seminars are needed. We believe this study’s initiatives will expand the discussion regarding efforts that can be made at the institutional and individual level to strengthen nurses’ ethical competence.

## Figures and Tables

**Table 1 ijerph-18-00241-t001:** Topics for ethics seminar.

Session	Topics	Contents	Data Collection
1	What is nursing ethics?	What is nursing ethics?	Respond to the questionnaire: Moral sensitivity, unethical behavior of nurses
		Why should a nurse be ethical?	
		What is bioethics?	
		Why should nurses know bioethics?	
2	What is moral thinking?	“Moral thinking is the process of rational reasoning about the contents of the value judgment of right and wrong” (lecture by a professor of ethics philosophy)	
3	Relational ethics in nursing organizations	Discussion of ethical and unethical situations experienced by nurses in nursing organizations	
4	Act on Decisions on Life-Sustaining Treatment for Patients in Hospice and Palliative Care or at the End of Life (Act No. 14013)	“Advanced care planning”	
		(lecture by a professor of nursing)	
5	A novice nurse’s ethical awareness vs. an experienced nurse’s ethical awareness	A nurse with less than one year of working experience and a nurse with ten years of experience present their concerns regarding ethical issues, and participants share their impressions of the presentation	
6	What does ethical nursing practice mean to me?	Participants shared their experience and thoughts on ethical nursing practice at the hospital	Second response to the questionnaire

**Table 2 ijerph-18-00241-t002:** Homogeneity of demographic characteristics, moral sensitivity, and unethical behavior.

General Characteristics		Experimental Group (*n* = 17)	Control Group (*n* = 18)	*χ*^2^, *t*, *z*	*p*
Age	Under 30	11 (64.7%)	10 (55.6%)	0.305	0.581
	30 or above	6 (35.3%)	8 (44.4%)		
Duration of working experience (years)	Under 3 years	3 (17.6%)	2 (17.6%)	0.315	0.854
	3–10 years	9 (52.9%)	9 (52.9%)		
	10 or above	5 (29.4%)	5 (29.4%)		
Experience of ethics education	No	14 (82.4%)	9 (50.0%)	4.062 ^a^	0.075
	Yes	3 (17.6%)	9 (50.0%)		
Moral sensitivity		36.65 (3.59)	37.56 (5.62)	−0.573 ^b^	0.571
Unethical behavior		12.59 (3.62)	10.67 (2.83)	−1.986 ^c^	0.075

^a^ Fisher’s exact test; ^b^
*t*-test; ^c^ Mann–Whitney U test; *z*-variables.

**Table 3 ijerph-18-00241-t003:** Correlations among the variables (*n* = 35) at post-test.

Variables	Moral Sensitivity	Unethical Behavior
Moral sensitivity	1	
Unethical behavior	−0.400 *	1

* *p* < 0.05.

**Table 4 ijerph-18-00241-t004:** Difference between pre and posttest in ethics seminar.

Variables	Group	Pre Test		Post Test		Mean Difference	*t*	*p*	Effect Size
		Mean	SD	Mean	SD				
Moral sensitivity	Experimental	36.65	3.587	38.29	5.687	−1.647	−1.039	0.314	0.252 ^a^
	Control	37.56	5.617	39.22	6.292	−1.667	−0.961	0.350	−0.227
Unethical behaviors	Experimental	12.59	3.624	9.47	1.663	3.118	3.363	0.004	0.816
	Control	10.67	2.828	10.22	3.282	0.444	0.714	0.485	0.168

^a^ Cohen’s d.

## Data Availability

The data presented in this study are available on request from the corresponding author. The data are not publicly available due to restrictions e.g., privacy or ethical.

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
