# Peer review of "Effect of Ethics Seminar on Moral Sensitivity and Ethical Behavior of Clinical Nurses"

_ijerph, 2020, doi:10.3390/ijerph18010241_

Round 1

Reviewer 1 Report

My training is in ethics, rather than social science; hence, I am not in the best position to make judgments about the statistical elements of the study, beyond the fact that they appear rational.   

It appears that in this study "ethical behavior" is largely reduced to "report/not report." If so, such a reduction may suggest that wrong behavior is behavior that is against hospital rules or written professional conduct standards, which is a limited perspective. It may suggest also that direct reporting is always the best response to the witnessing of an infraction, which I expect is false. Nurses must also have an ethical sensitivity to the ethical climate of their unit and hospital, and recognize the need to foster trust and confidence among one's peers. There are multiple ethical values to balance.

I appreciated the relative detail regarding the content of the ethics seminar.

In editing, take care to correct mismatches between nouns (sing/plural) and verbs, as well as a few missing articles. Otherwise the English is clear and elegant. A pleasure to read. 

Author Response

Reviewer 1. comments

My training is in ethics, rather than social science; hence, I am not in the best position to make judgments about the statistical elements of the study, beyond the fact that they appear rational. 

It appears that in this study "ethical behavior" is largely reduced to "report/not report." If so, such a reduction may suggest that wrong behavior is behavior that is against hospital rules or written professional conduct standards, which is a limited perspective. It may suggest also that direct reporting is always the best response to the witnessing of an infraction, which I expect is false. Nurses must also have an ethical sensitivity to the ethical climate of their unit and hospital, and recognize the need to foster trust and confidence among one's peers. There are multiple ethical values to balance.

I appreciated the relative detail regarding the content of the ethics seminar.

In editing, take care to correct mismatches between nouns (sing/plural) and verbs, as well as a few missing articles. Otherwise the English is clear and elegant. A pleasure to read.

----- Thank you for your review and valuable comments. An expert in www.editage.com has edited throughout this paper.

Reviewer 2 Report

Dear authors, congratulations on the subject of study of the manuscript, the training in ethical values ​​is studied, an important aspect for the daily work performance of the nursing staff.

After reviewing the manuscript, I submit the following comments.

Best regards,

The abstract section

In lines 16-17 "After the ethics seminar, the experimental group's moral sensitivity was significantly increased (t = -1.039, p = 0.314).", Although we can accept the increase, but it cannot be considered as important data as it is not statistically significant (p is greater than 0.05) and does not provide any additional data, especially when they specify on line 118, "p <.05 was considered statistically significant with a 95% confidence interval". So it is important to correct that assertion.

Taking into account the previous comment, in lines 20-21 “We conclude that ethics seminars can 20 enhance moral sensitivity” does not correspond to the data provided as it is not statistically significant.

The introduction section

On line 46 “… [8, pp. 23–24]… ”should modify it, putting the consulted pages in the corresponding bibliographic reference, to unify the criterion of bibliographic references.

The materials and methods section

You must rename this section following the scheme that is on the publisher's website.

In the 2.1 Research Design subsection

You should comment on whether the control group that does not receive the ethics seminars within the study (although it is offered later), would be given the pretest-posttest to compare results, since it is not clear, since they have not been carried out a intervention.

The 2.2 Participants subsection

In lines 92-96, “The authors posted notices of recruitment for research participants on the hospital bulletin board, but most participants were recruited through snowball sampling, that is, some participants introduced their acquaintances to the study. As participants were recruited, they were divided into a control group and an experimental group ”should be included in this subsection. In addition, you must specify how you included each of the participants in each of the groups, randomly or with another procedure.

The 2.4 Data Collection subsection

In lines 92-96, "The study was conducted in accordance with the Declaration of Helsinki and was approved by 91 the Institutional Review Board of Chung-Ang University Hospital (1041078-201803)." It is recommended that any ethical aspect of the study should be incorporated into a section on it.

The 2.5 Data Analysis

In the lines 114-115 “As a result of the Kolmogorov-Smirnov and Shapiro-Wilk tests, t-test was used to compare differences between the two groups because the dependent variable was normally distributed”. You must specify which of the two tests you took into consideration to determine whether to perform a parametric or non-parametric test, depending on the size of the samples. As I do not have access to the raw statistical results, it is not appreciated if there is a divergence between the application of both tests.
Likewise, although Pearson's correlation is used to determine if there is a relationship between two quantitative variables, they do not make this point clear and with which variables. You must specify them to clarify this point.
In lines 117-118 "The effect of ethical seminars on moral sensitivity and unethical behavior was analyzed using paired t-test", you do not make it clear if they apply it to both groups or only to the control group, in the case of no pretest and post-test in the control group, in relation to the comments In the 2.1 Research Design subsection

Results and discussion section

In the lines 135-136 “After the ethics seminar, the experimental group's moral sensitivity was significantly increased (t = -1.039, p = 0.314).”, In the lines 163-165 “After attending the ethics seminar, the experimental group's moral sensitivity score increased. In a previous study [15], ethics education for nursing college students effectively increased moral sensitivity ”, as I have commented previously, as it does not have a p lower than 0.05, it cannot be taken as a relevant result in the study. So it is important to correct that assertion.

The conclusion section

As in previous sections, it is necessary to modify what refers to the results on moral sensitivity

The References section:

It can be seen that the bibliographic references 1 and 5 are poorly formulated.

Author Response

Dear authors, congratulations on the subject of study of the manuscript, the training in ethical values ​​is studied, an important aspect for the daily work performance of the nursing staff.

After reviewing the manuscript, I submit the following comments.

Best regards,

The abstract section

In lines 16-17 "After the ethics seminar, the experimental group's moral sensitivity was significantly increased (t = -1.039, p = 0.314).", Although we can accept the increase, but it cannot be considered as important data as it is not statistically significant (p is greater than 0.05) and does not provide any additional data, especially when they specify on line 118, "p <.05 was considered statistically significant with a 95% confidence interval". So it is important to correct that assertion.

----- As the reviewer pointed out, there was an error when we described the statistical results. Moral sensitivity did not increase significantly. We revised the sentence.

Taking into account the previous comment, in lines 20-21 “We conclude that ethics seminars can 20 enhance moral sensitivity” does not correspond to the data provided as it is not statistically significant.

----- We revised the sentence as follows: We conclude that ethics seminars can reduce the risk of unethical behavior among nurses(on line 21 in the abstract).  

The introduction section

On line 46 “… [8, pp. 23–24]… ”should modify it, putting the consulted pages in the corresponding bibliographic reference, to unify the criterion of bibliographic references.

----- We added the pages to the corresponding reference in the reference session (on line 238).

The materials and methods section

You must rename this section following the scheme that is on the publisher's website.

----- "2. Methods" was changed to "2. Material and Methods".

In the 2.1 Research Design subsection

You should comment on whether the control group that does not receive the ethics seminars within the study (although it is offered later), would be given the pretest-posttest to compare results, since it is not clear, since they have not been carried out a intervention.

----- We added The control group was not exposed to the ethics seminar during the course of the study.” (on lines 64-65).

The 2.2 Participants subsection

In lines 92-96, “The authors posted notices of recruitment for research participants on the hospital bulletin board, but most participants were recruited through snowball sampling, that is, some participants introduced their acquaintances to the study. As participants were recruited, they were divided into a control group and an experimental group ”should be included in this subsection. In addition, you must specify how you included each of the participants in each of the groups, randomly or with another procedure.

----- We moved the sentences into the 2.2 Participants subsession and speified the procedure to divide the groups: The authors posted recruitment notices for research participants on the hospital bulletin board; however, most participants were recruited through snowball sampling. The recruited participants were randomly divided into the control group and the experimental group. Despite this, some participants joined either the experimental group or the control group at their own request.”(on lines 71-74, page 2)

The 2.4 Data Collection subsection

In lines 92-96, "The study was conducted in accordance with the Declaration of Helsinki and was approved by 91 the Institutional Review Board of Chung-Ang University Hospital (1041078-201803)." It is recommended that any ethical aspect of the study should be incorporated into a section on it.

----- We rewrote some ethical aspect of the study: “The study was conducted in accordance with the Declaration of Helsinki and was approved by the Institutional Review Board of Chung-Ang University Hospital (1041078-201803). The authors made sure that participants understood the purpose and procedure of this study and explained that they could withdraw at any time if they no longer wished to participate. All participants completed a written consent form voluntarily ~” (on lines 96-100).

The 2.5 Data Analysis

In the lines 114-115 “As a result of the Kolmogorov-Smirnov and Shapiro-Wilk tests, t-test was used to compare differences between the two groups because the dependent variable was normally distributed”. You must specify which of the two tests you took into consideration to determine whether to perform a parametric or non-parametric test, depending on the size of the samples. As I do not have access to the raw statistical results, it is not appreciated if there is a divergence between the application of both tests.

----- We revised the sentences as follows: The χ2-test and Fisher's exact test were used for the homogeneity test between the two groups. For the homogeneity test for both groups' moral sensitivity and unethical behavior, the Shapiro-Wilk test was used to test for normality. If normality was not satisfied (e.g., the control group's unethical behavior), it was calculated using the Mann-Whitney test. The correlations between variables (moral sensitivity and unethical behavior) were analyzed using Pearson correlation analysis. In analyzing the differences between groups to compare the effects of the ethics seminar on moral sensitivity and unethical behavior, the Shapiro-Wilk test indicated that data were normally distributed. Thus, a paired t-test was used for both groups. 

Likewise, although Pearson's correlation is used to determine if there is a relationship between two quantitative variables, they do not make this point clear and with which variables. You must specify them to clarify this point.

--- We specified the variables (moral sensitivity and unethical behavior) on line 118.

In lines 117-118 "The effect of ethical seminars on moral sensitivity and unethical behavior was analyzed using paired t-test", you do not make it clear if they apply it to both groups or only to the control group, in the case of no pretest and post-test in the control group, in relation to the comments In the 2.1 Research Design subsection

--- In analyzing the differences between groups to compare the effects of the ethics seminar on moral sensitivity and unethical behavior, the Shapiro-Wilk test indicated that data were normally distributed. Thus, a paired t-test was used for both groups (on line 119-121).

Results and discussion section

In the lines 135-136 “After the ethics seminar, the experimental group's moral sensitivity was significantly increased (t = -1.039, p = 0.314).”, In the lines 163-165 “After attending the ethics seminar, the experimental group's moral sensitivity score increased. In a previous study [15], ethics education for nursing college students effectively increased moral sensitivity ”, as I have commented previously, as it does not have a p lower than 0.05, it cannot be taken as a relevant result in the study. So it is important to correct that assertion.

----- We have corrected the mistake in interpreting statistical results and revised the corresponding sentences in the discussion session.

“Although the experimental group’s moral sensitivity score did not significantly increase after attending the ethics seminar, previous studies have found that ethics education for nursing college students effectively increased their moral sensitivity [15]. In this study, the ethics seminar was held only once a month; thus, this may not have been enough to increase moral sensitivity.” (lines 172-175, page 5)

The conclusion section

As in previous sections, it is necessary to modify what refers to the results on moral sensitivity

The References section:

It can be seen that the bibliographic references 1 and 5 are poorly formulated.

----- When we searched the web site [1] again, the data was not available, so we cited another reference. We confirmed that the website corresponding to [5] was available.

Thank you so much for taking precious time to review our paper. Thanks to you, the quality of the paper has improved. Thank you.

Round 2

Reviewer 2 Report

Dear authors, I have appreciated an improvement in your revised manuscript.

You have responded to all the proposals for improvement that have been suggested by me and it only remains to congratulate you.

Best regards,